# Lactobacilli as a Vector for Delivery of Nanobodies against Norovirus Infection

**DOI:** 10.3390/pharmaceutics15010063

**Published:** 2022-12-25

**Authors:** Yoshikazu Yuki, Fanglei Zuo, Shiho Kurokawa, Yohei Uchida, Shintaro Sato, Naomi Sakon, Lennart Hammarström, Hiroshi Kiyono, Harold Marcotte

**Affiliations:** 1Division of Mucosal Immunology, IMSUT Distinguished Professor Unit, The Institute of Medical Science, The University of Tokyo, Tokyo 108-8639, Japan; 2HanaVax Inc., Chiba 260-0856, Japan; 3Department of Human Mucosal Vaccinology, Chiba University Hospital, Chiba 260-8670, Japan; 4Department of Biosciences and Nutrition, Karolinska Institutet, SE-141 83 Huddinge, Sweden; 5Department of Microbiology and Immunology, School of Pharmaceutical Sciences, Wakayama Medical University, Wakayama 641-0011, Japan; 6Department of Virology, Research Institute for Microbial Diseases, Osaka University, Osaka 565-0871, Japan; 7Department of Microbiology, Osaka Institute of Public Health, Osaka 537-0025, Japan; 8Mucosal Immunology and Allergy Therapeutics, Institute for Global Prominent Research, Future Medicine Education and Research Organization, Chiba University, Chiba 263-8522, Japan; 9CU-UCSD Center for Mucosal Immunology, Allergy, and Vaccines (cMAV) Division of Gastroenterology, Department of Medicine, University of California, San Diego, CA 92093-0956, USA

**Keywords:** single-domain antibody, nanobody, *Lactobacillus*, norovirus

## Abstract

Passive administration of neutralizing antibodies (Abs) is an attractive strategy for the control of gastrointestinal infections. However, an unanswered practical concern is the need to assure the stability of sufficient amounts of orally administered neutralizing Abs against intestinal pathogens (e.g., norovirus) in the harsh environment of the gastrointestinal tract. To this end, we expressed a single-domain Ab (VHH, nanobody) against norovirus on the cell surface of *Lactobacillus*, a natural and beneficial commensal component of the gut microbiome. First, we used intestinal epithelial cells generated from human induced pluripotent stem cells to confirm that VHH 1E4 showed neutralizing activity against GII.17 norovirus. We then expressed VHH 1E4 as a cell-wall–anchored form in *Lactobacillus paracasei* BL23. Flow cytometry confirmed the expression of VHH 1E4 on the surface of lactobacilli, and *L. paracasei* that expressed VHH 1E4 inhibited the replication of GII.17 norovirus in vitro. We then orally administered VHH 1E4-expressing *L. paracasei* BL23 to germ-free BALB/c mice and confirmed the presence of lactobacilli with neutralizing activity in the intestine for at least 10 days after administration. Thus, cell-wall-anchored VHH-displaying lactobacilli are attractive oral nanobody deliver vectors for passive immunization against norovirus infection.

## 1. Introduction

Noroviruses are non-enveloped RNA viruses that are divided into seven genogroups according to their capsid sequences. The 28 genotypes of viruses in the GI and GII genogroups can infect humans causing brutal illness [1]. Human norovirus causes an estimated 200,000 deaths annually worldwide and is a common infection in both developed and developing countries in children younger than five years, the elderly, and immunocompromised people [2]. During the past fifteen years, GII.4 noroviruses have been the major viruses worldwide, but those of the GII.17 genotype have recently become the predominant strains in southeast Asia, including Japan [3]. Currently, no licensed norovirus vaccines or drugs are available to control severe gastrointestinal infectious diseases caused by this pathogen. In addition, even though two vaccines against norovirus GI.1 and GII.4 are in development [4,5,6], a complementary strategy of immune therapy may be necessary when vaccination alone is insufficiently effective.

The variable fragment of llama heavy chain antibody (VHH) is a small (15 kDa), heat- and acid-stable protein with higher solubility than, but similar affinities as, conventional antibodies [7]. Accordingly, VHHs have become attractive candidates for oral anti-noroviral therapy and prophylaxis against norovirus infection. Regarding the safety of VHHs, the first VHH-based medicine, Cablivi (caplacizumab), has recently been approved in Europe and the United States for use as a subcutaneous injection in adults with acquired thrombotic thrombocytopenic purpura [8]. In addition, in a Phase II clinical trial in Bangladesh, oral administration of yeast expressing a VHH (named ARP1) to rotavirus effectively reduced the severity of diarrhea in children [9]. In that clinical study, a daily VHH dose of 15–30 mg/kg had to be administered for five consecutive days to achieve the reported effect. To support the use of VHH, we previously developed a rice-based expression system, which produced high yields of VHHs (0.5–1.5% of rice seed [w/w]) [10,11,12]. We also developed an expression system in which VHHs are expressed as a cell-surface–anchored form in *lactobacilli* [13,14,15]; we have now two unique oral nanoantibody delivery vehicles using *lactobacillus* and MucoRice systems. In addition, oral inoculation with lactobacilli producing a VHH (ARP1) against rotavirus was highly effective in reducing disease duration, disease severity, and viral load in a mouse model [13].

In the current study, we developed *Lactobacillus paracasei* BL23 cells that display a VHH against GII.17 norovirus as a potential means to control norovirus infections. We used inducible pluripotent stem cell (iPSC)-derived human intestinal epithelial cells (IECs) [16] to confirm the neutralization activity of the VHH-bearing lactobacilli. Furthermore, we showed that orally administered lactobacilli displaying GII.17 VHH demonstrated neutralizing activity in the intestines of germ-free mice. Together, our findings suggest that *Lactobacillus* displaying cell-wall-anchored VHH is another attractive oral delivery vehicle for passive immunization against norovirus infection.

## 2. Materials and Methods

### 2.1. Bacterial Strains, Plasmids, and Growth Conditions

The plasmids and bacterial strains used in this study are listed in Appendix A [17]. *Lactobacillus paracasei* BL23 (previously known as *L. casei* or *L. zeae* ATCC 393 pLZ15^−^) cells were grown anaerobically at 37 °C on MRS agar plates (Difco, Becton Dickinson, Sparks, MD, USA) or in MRS broth without shaking. *Escherichia coli* DH5α cells (Invitrogen, Carlsbad, CA, USA) were grown at 37 °C on Luria–Bertani (LB) plates or in LB broth with 220 rpm orbital shaking. The VHH fragment was expressed in *Lactobacillus* under the control of the aggregation-promoting factor (APF) promoter in the *Lactobacillus* expression vector pAF900 [18]. When appropriate, ampicillin (100 µg/mL) or erythromycin (300 µg/mL) was used for *E. coli* transformants and erythromycin (5 μg/mL) was used for *L. paracasei* transformants.

### 2.2. Construction of Recombinant L. paracasei BL23 Expressing VHH 1E4

The VHH fragment 1E4 against human norovirus GII.17 (Kawasaki 308) was generated as described previously [18] The corresponding gene, *1E4*, was codon-optimized according to the synonymous codon usage of *L. paracasei* and synthesized (GeneScript, Rijswijk, The Netherlands). The gene encoding 1E4 was cloned into the *Lactobacillus* expression vector pAF900, which mediates the display of the protein on the cell surface via its fusion to an LPXTG cell-wall anchor domain (i.e., the last 243 C-terminal amino acids of the proteinase prtP of *L. paracasei* BL23) [19]. The synthesized DNA fragment contained a SalI restriction site followed by the APF promoter, APF signal peptide, VHH gene, and a NotI restriction site, hence fusing the VHH gene directly to the signal peptide. The synthetic gene was excised from the pUC57 plasmid by using SalI and NotI Fast Digest restriction enzymes (Thermo Scientific, Dreieich, Germany) and ligated into SalI-NotI-digested pAF900 [18]. The resulting plasmid, pAF900-1E4, was used to transform *E. coli* DH5α cells. Plasmid was isolated from respective clones by using a Qiagen Plasmid Midi Kit (Qiagen, Hilden, Germany) and sequence verified. The pAF900-1E4 plasmid (and the empty vector pIAV7) were used to transform *L. paracasei* BL23 [20] via electroporation as described previously [21], thus generating *L. paracasei* BL23-pIAV7 and BL23-1E4. The recombinant *L. paracasei* BL23 strains were verified through PCR analysis and sequencing.

### 2.3. Western Blot Analysis

Two ml of an MRS growth culture from *L. paracasei* BL23 transformants were centrifuged at 6000× *g* for 5 min. The culture supernatants and cell extracts were run on a 12% SDS–polyacrylamide gels and transferred onto a nitrocellulose membrane (GE Healthcare, Amersham, UK) as previously described [22]. The membrane was blocked with 5% (*w*/*v*) powdered skim milk in PBS containing 0.05% [*v*/*v*] Tween 20 and successively incubated with mouse anti-E-tag antibody (1 µg/mL; Phadia AB, Uppsala, Sweden) (2 h, room temperature) and horseradish peroxidase-labeled goat anti-mouse antibody (DAKO A/S, Glostrup, Denmark) (1 h, room temperature). Proteins were detected by using the ECL Plus Western Blotting Detection System (GE Healthcare) [22]. 

### 2.4. ELISA Using Whole Bacteria in Suspension to Confirm Cell-Surface Display of VHH

For a semi-quantitative assay to verify surface expression of the VHH, *L. paracasei* BL23 transformants were grown in MRS broth containing 5 µg/mL erythromycin until the cultures reached an OD_600_ of 1.0. Samples of the bacterial cultures (400 µL each) were centrifuged. The resulting cell pellets were washed twice with PBS, resuspended in 400 µL PBS containing 1% bovine serum albumin. The cells in 100-µL quantities of this suspension were pelleted again, and the resulting pellet was resuspended in 100 µL PBS containing 1% bovine serum albumin and 1 µg/mL mouse anti-E-tag antibody and incubated on ice for 30 min. After one wash with PBS, the cells were resuspended in 100 µL of buffer containing alkaline phosphatase-conjugated rabbit anti-mouse IgG (1 µg/mL) and then incubated on ice for 30 min. After two washes with PBS, the cells were suspended in 200 µL diethanolamine buffer; 100 µL of bacterial suspension from each clone was added to a well of a 96-well microtiter plate, followed by 100 µL of diethanolamine buffer containing 1 mg/mL p-nitrophenyl phosphate (Sigma–Aldrich, Darmstadt, Germany). Plates were incubated for 10–30 min at room temperature, after which the absorbance at 405 nm was read in a Varioskan Flash multimode reader (Thermo Scientific). *Lactobacillus paracasei* cells containing the empty plasmid (pIAV7) were used as a negative control.

### 2.5. Plasmid Stability According to Plate Count Assay

*Lactobacillus paracasei* BL23 transformants were grown in MRS broth containing 5 µg/mL erythromycin until the cultures reached an OD_600_ of 1.0. The cultures were serially diluted in PBS and plated on MRS plates supplemented with or without 5 µg/mL erythromycin. The plates were incubated at 37 °C for 2 days. The percentage of plasmid persistence was calculated as A1/A0 × 100%, where A1 and A0 are the viable counts of the antibiotic plate and antibiotic-free plate, respectively.

### 2.6. Neutralization Assay of Human Norovirus

The *L. paracasei* strains BL23-pIAV7 and BL23-1E4 were grown in MRS broth containing 5 µg/mL erythromycin in an anaerobic system (BBL GasPak Anaerobic Systems, BD BBL, Sparks, MD, USA) until the cultures reached an OD_600_ of 1.0 and then diluted in buffer to 3 × 10^6^ cells per 50 μL. Neutralization assay of human noroviruses was performed as described previously [18]. Human iPSC-derived IECs were cultured and maintained in Matrigel (Corning, Tewksbury, MA, USA), and monolayers of IECs were prepared as previously described [16]. Human norovirus GII.17 Kawasaki 308 (strain Hu/GII.17/OsakaFB16421/2017/JP) was diluted to 5.0 × 10^6^ genome equivalents per 100 µL with and without BL23-pIAV7 or BL23-1E4 (3 × 10^6^ cells) in base medium [16] containing 5 μg/mL erythromycin and then incubated for 90 min. Wells of prepared IECs (3 to 6 wells per sample) were inoculated with 100 µL of diluted virus suspension with or without lactobacilli and incubated for 3 h at 5% CO_2_ and 37 °C. The inoculum was then removed, and the IECs were washed three times with 150 µL of base medium. We then added 100 μL of differentiation medium [16] containing 5 μg/mL erythromycin and 0.03%, gently pipetted up and down twice, spun down the debris, and collected the medium. These steps were repeated, and the suspensions were pooled (total, 200 µL) and collected as 1- or 3 h post-infection reference samples. Another 100 µL of differentiation medium containing 0.03% bile was added to each well, and the mixtures were then incubated for 48 h in a 5% CO_2_ incubator at 37 °C. The supernatants were collected, with one wash, in the same way as the reference samples (total, 200 µL). The inhibitory activity of *L. paracasei* BL23-pIAV7 or BL23-1E4 against norovirus replication was shown as the percentage inhibition relative to untreated *L. paracasei* BL23-pIAV7 or BL23-1E4. Recombinant VHH 1E4 was expressed in E. coli system and purified by the method described previously, and 5 μg of 1E4 was used as a positive control [18]. 

### 2.7. Oral Administration, Isolation, and Culture of L. paracasei BL23 Strains

Germ-free female BALB/c mice (4–7 weeks old) were purchased from Japan SLC and used for oral administration of *L. paracasei* BL23-pIAV7 or BL23-1E4 at the Institute of Medical Science of The University of Tokyo. The animal study protocol was approved by the Animal Care and Use Committee of The University of Tokyo (protocol code PA21-64 and approval date 23 March 2022).

*Lactobacillus paracasei* BL23-pIAV7 or BL23-1E4 was grown at 37 °C for 15 h in MRS broth containing 5 µg/mL erythromycin in an anaerobic system (BBL GasPak Anaerobic Systems, BD BBL) without shaking until the culture reached an OD_600_ of 1.0. The bacteria were counted, and doses of *L. paracasei* BL23-pIAV7 or BL23-1E4 (1 × 10^9^ cells per 200 mL) were administered orally to germ-free mice (*n* = 3). 

For flow cytometry and neutralization assay of human norovirus, feces from germ-free mice were collected at 4, 8, 24, 48, 72, 120, 240, and 336 h (2 weeks) after oral administration of *L. paracasei* BL23-pIAV7 or BL23-1E4; fecal pellets were weighed and added to PBS to achieve 100 mg feces per milliliter of PBS. These mixtures were then diluted 500-fold and 2000-fold by using PBS, and aliquots were spread on MRS agar plates (MRS broth containing 1% CaCO_3_, 1.6% agar, and 5 µg/mL erythromycin); plates were incubated at 37 °C for 48 h in an anaerobic system (BBL GasPak Anaerobic System, BD BBL) without shaking. Single colonies of *L. paracasei* were picked and then grown at 37 °C for 15 h in MRS broth containing 5 µg/mL erythromycin in an anaerobic system (BBL GasPak Anaerobic Systems) without shaking until the cultures reached an OD_600_ of 1.0. Bacteria were counted, and *L. paracasei* BL23-pIAV7 or BL23-1E4 (3 × 10^6^ cells per 50 μL) underwent flow cytometry and neutralization assay of human norovirus.

### 2.8. Flow Cytometric Analysis of Lactobacilli

For flow cytometry, bacteria were washed with PBS, and 5 × 10^7^ cells were transferred into a 2-mL tube. A total of 2 μL rabbit anti-E tag antibody (Abcam, Cambridge, UK) was added, and cells were incubated for 30 min at 37 °C. After two wash with PBS, the cells were resuspended in 100 µL PBS containing 2 μL of Brilliant Violet 421-labelled goat anti-rabbit IgG (Biolegend, San Diego, CA, USA) and then incubated for 30 min at 37 °C. The cells were washed twice with PBS and then fixed in 1 mL of 1% formaldehyde (Sigma–Aldrich) at room temperature. A total of 12.5 μL thiazole orange (Biotium, Hayward, CA, USA) was added before bacterial cells were analyzed in a Attune NxT flow cytometer (Thermo Fisher Scientific). The number of live *L. paracasei* BL23 in the gut of germ-free BALB/c mice after oral administration was calculated as the percentage of E-tag-positive cells per thiazole orange-positive cells.

### 2.9. Statistical Analysis

Results were compared using unpaired two-tailed Student’s *t*-tests (Prism 7, GraphPad Software).

## 3. Results

### 3.1. Production of 1E4-VHH-Displaying Lactobacilli

We directly fused VHH 1E4 to the signal peptide, leaving one alanine residue at the N-terminal for correct cleavage of the signal peptide (Figure 1a). We constructed the expression cassette (pAF900-1E4, expected size 40.04 KDa) so that 1E4 would be expressed as a protein anchored on the cell surface via fusion to the prtP anchor sequence.

After transformation into *L. paracasei* BL23 cells, expression of VHH 1E4 was verified by Western blot analysis of the culture supernatant and cell fractions (Figure 1b). VHH expressed by lactobacilli as an anchored form was present at the expected size (40 KDa) and in the expected cell fraction of *L. paracasei* BL23-1E4. The presence of higher molecular-weight bands is most likely due to peptidoglycan subunits covalently linked to the recombinant protein, which retard its migration, whereas the bands at lower molecular weights are degradation products within the bacterial cells due to overexpression of the recombinant protein [23]. 

The display of VHH 1E4 on *L. paracasei* BL23 cell surfaces was confirmed through a fluid-based assay using a mouse monoclonal anti-E-tag primary antibody and an alkaline phosphatase-conjugated rabbit anti-mouse secondary antibody. Compared with the non-expressing *L. paracasei* strain, the VHH 1E4-expressing cells gave a strong signal, confirming the expression of VHH 1E4 on the surface of the *Lactobacillus* cells (Figure 1c). The plasmid stability and persistence of the recombinant *L. paracasei* BL23 strain expressing VHH 1E4 were measured through colony counts. For both the empty vector and the strain carrying the expression cassette, the proportion of plasmid-containing cells declined by 20% after approximately 6–8 h of culture in MRS broth (Figure 1d).

### 3.2. Norovirus Neutralization by L. paracasei BL23 Expressing VHH 1E4 Using an In Vitro Propagation Assay with Human iPSC-Derived IECs

Next, we used human iPSC-derived IECs to test whether *L. paracasei* BL-23 cells expressing 1E4 VHH neutralized the propagation of human norovirus GII.17. Because of the prolonged culture time (48 h) needed for assessment of neutralization activity by the VHH 1E4-expressing recombinant *L. paracasei* BL23 strain, we changed the antibiotic treatment in the in vitro propagation assay using human iPSC-derived IECs to erythromycin instead of the penicillin–streptomycin originally described. GII.17 norovirus has the ability to survive in low pH (2% citric acid, pH 3–5) and replicates in iPSC-derived IECs [24]. Although the growth of *L. paracasei* BL23 pIAV7 (the negative control strain) might result in acidic pH in the cell culture, there was no pH-effect on GII.17 norovirus propagation in human iPSC-derived IEC in vitro when compared with that of PBS (Figure 2). Under these conditions, a dose of 3 × 10^6^
*L. paracasei* BL23-1E4 cells neutralized 2 × 10^6^ genome equivalents of norovirus GII.17 (Kawasaki 308); this dosage was equivalent to 5 μg of recombinant 1E4 (Figure 2). 

### 3.3. In Vitro Norovirus Neutralization by VHH-Displaying Lactobacillus Cells Isolated from the Feces of Germ-Free Mice after Their Oral Administration

Because GII.4 norovirus does not replicate in the intestine of any animal species and no standard animal model for norovirus infection is currently available, we used germ-free mice to investigate whether *L. paracasei* BL23-1E4 could survive and display VHH 1E4 in the environment of the gastrointestinal (or in vivo) environment after oral delivery. To isolate *L. paracasei* BL23 from feces, we collected feces from inoculated mice from 4 to 340 h for a total of 2 weeks after oral administration and then cultured the samples anaerobically on MRS plates containing erythromycin. This assay confirmed the presence of *L. paracasei* BL23 in the feces of germ-free mice for as long as 240 h (10 days) after oral administration (Appendix A). In addition, flow cytometric analysis confirmed the display of VHH on the surface of *L. paracasei* BL23-1E4 cells isolated from the feces of orally inoculated mice (Figure 3). Furthermore, unlike the negative control strain (BL23-pIAV7), VHH-displaying *Lactobacillus* BL23-1E4 cells that were recovered from the feces of inoculated mice neutralized human norovirus GII.17 propagation in vitro (Figure 4). These combined results showed that orally administered VHH-displaying *Lactobacillus* survive in the gastrointestinal tract, appropriately display VHH on their cell surface, and effectively neutralize norovirus in an in vitro assay, suggesting that VHH-displaying *Lactobacillus* are a potential anti-norovirus treatment. 

## 4. Discussion

Approximately 200,000 people die annually worldwide due to norovirus infection [2]; this figure translates to an economic burden of $60.3 billion globally in social costs due to this disease in addition to $4.2 billion in direct healthcare costs each year [25]. Vaccines against the GII.4 and GI.1 strains are currently under development [5,6], but additional strategies involving passive immunity may be needed. In a clinical study in Bangladesh, daily oral dosage of yeast-based VHH (15–30 mg/kg) had to be continued for 1 week to control rotaviral disease in children [9]. To obtain the large quantities of VHH needed, we previously developed oral antibody-producing rice (MucoRice-VHH) for passive immune therapy against noroviral infection [11,12]. Rice-based VHH is cold-chain-free, and although a rice-based system can produce large amounts of VHH, developing such a system is time consuming. Here, we developed oral antibody (VHH, nanobody)-displaying *L. paracasei* BL23 as an option for passive immunotherapy to protect against and treat noroviral infections in healthy persons of all ages and in various immunocompromised populations.

In a mouse model of rotavirus-induced diarrhea, oral administration of *L. paracasei* strains that expressed cell-surface-anchored forms of VHH more effectively suppressed disease severity and viral load than did those that secreted VHHs [13]. Given those findings, we constructed an *L. paracasei* strain that expressed cell surface-anchored VHH 1E4 and confirmed that it appropriately expressed and displayed the VHH. We then assessed the plasmid stability and persistence of *L. paracasei* BL23 strains that expressed VHH 1E4 and found that, under antibiotic selection, about 80% of cells expressed and displayed the VHH. Similar results regarding expression have been obtained with other VHHs [14,15]. In *Lactobacillus*, decreases in the expression of membrane-anchored proteins frequently are due to plasmid instability, and the expression cassettes should be integrated into the *Lactobacillus* genome to stabilize expression [14,15]. We previously showed that *L. paracasei* BL23 producing surface-anchored ARP1, engineered by using either a plasmid or integration system, conferred similar protection in a mouse model of rotavirus infection, thus suggesting the feasibility of using a chromosomally integrated expression system for the delivery of VHH against norovirus [19].

A lack of well-characterized in vitro and in vivo infection models has limited the development of human norovirus research. Although gnotobiotic piglet and pigtail macaque models of human norovirus infection have been reported, no standard animal models have yet been established [26]. A recent breakthrough in human norovirus research is the development of an in vitro culture system using human intestinal enteroid cells derived from biopsy tissue collected from adults [27]. In this regard, we have developed a propagation system for human noroviruses that uses human iPSC-derived IECs [16]. Human norovirus infects by attaching human histo-blood group antigens as co-receptors [28]; the primary receptor(s) for noroviral infection of host cells are currently unknown. Although human primary IECs, including iPSC-derived IECs, express histo-blood group antigens, norovirus replication typically also requires supplementation with bile, which contains unidentified components. In particular, GI.1, GII.3, and GII.17, but not GII.4, noroviruses require bile [16,27]. Despite the use of bile, we think that the efficiency of norovirus replication in enteroid models including iPSC-derived human IECs is low compared with that of the in vivo human intestinal environment. Therefore, although the enteroid model does not completely mimic the human intestine, it remains effective as a neutralization assay. By using human IECs, we previously found that the cross-reactivity of VHHs against VLP of norovirus GII genotypes did not correlate with cross-neutralization activity and that there was no universal VHH for neutralization among GII norovirus genotypes. For example, VHH 1E4 neutralizes GII.17 norovirus but not other GII genotypes [18]. Therefore, genotype-specific VHHs, including those for GII.2, GII.4, and GII.17 noroviruses, need to be developed. 

Using our iPSC-derived IECs GII.17 norovirus culture system, we confirmed the neutralization activity of lactobacilli-based norovirus VHH 1E4. According to flow cytometry using calibrated fluorescent microspheres, roughly 1000 VHH were displayed on the surface of each bacterial cell. Thus, 3 × 10^6^
*L. paracasei* BL23-1E4, which completely neutralized 2 × 10^6^ genome equivalents of norovirus GII.17, contains approximately 3 × 10^9^ VHH molecules on the cell surface. According to our previous paper [18], 0.05 to 5 mg of VHH 1E4 inhibited 2 × 10^6^ genome equivalents of norovirus GII.17, yielding approximately 2 × 10^12^ to 2 × 10^14^ VHH molecules [29]. Therefore, 1E4 VHH-displaying *L. paracasei* was at least 1000 times more effective at neutralizing norovirus than was free VHH 1E4. The numerous antibody fragments expressed on the bacterial surface result in the formation of ‘biological beads’ that afforded high-avidity binding due to multivalency and, thus, most likely promoted strong agglutination and subsequent neutralization of the virus.

We also confirmed that orally administered VHH-displaying *L. paracasei* BL23-1E4 survived in the intestine of germ-free mice and continued to present VHH on the surface of *L. paracasei* cells and that VHH-displaying *Lactobacillus* BL23-1E4 cells that were isolated from the feces of the inoculated mice neutralized human norovirus in vitro. Because this assay involved a second round of culture on erythromycin-containing MRS plates, these results suggest that orally administered VHH-displaying *L. paracasei* BL23-1E4 was stable in the gastrointestinal tract and therefore had the potential to suppress norovirus infection. Together, our findings indicate that oral administration of nanobody-displaying *L. paracasei* BL23-1E4 may be effective for prophylaxis against and treatment of GII.17 noroviral infections.

It is more likely that *L. paracasei* BL23-1E4 would be used as a prophylactic when outbreaks of norovirus infection occur. However, lactobacilli are, in general, unlikely to persist long-term in the human intestine; for example, the probiotic strain *L. rhamnosus* GG remained for only about 1 week after oral administration was discontinued 12 days previously [30]. Therefore, daily or weekly repeated oral administration of *L. paracasei* BL23-1E4 will likely be necessary, particularly during norovirus seasons. The clinical utility of the plasmid-based VHH-displaying lactobacilli we developed in the current study will benefit from not only improved plasmid stability but also strategies to prevent environmental contamination due to the administered organisms. In terms of their development as pharmaceuticals, VHH-displaying *Lactobacillus* strains are genetically modified organisms, and we need to prevent or minimize their unintended release into the environment. In general, orally administered lactobacilli transiently colonize the gastrointestinal tract for a maximum of approximately 1 month [31]. In our current study, *L. paracasei* BL23-1E4 were present in the feces of germ-free mice for at least 10 days after inoculation (Appendix A); in comparison, *Lactobacillus* that displayed a VHH against rotavirus survived for only 2 to 4 days after oral treatment of wild-type mice [13]. Therefore, we think that following oral administration in germ-free mice, *L. paracasei* BL23-1E4 will be eliminated after around two weeks most likely due to loss of plasmid. To mitigate the likelihood of environmental contamination, our group recently developed a system that couples chromosomal integration of the expression cassette with marker-free selection, which we call ‘inducible plasmid self-destruction’ [32]. This new genome-editing tool broadens the potential use of genetically modified organisms for medical drug application and currently is being used to engineer *Lactobacillus* that display norovirus-specific VHH for passive immunization in both the therapeutic and prophylactic settings. 

## 5. Conclusions

We developed a nanobody-displaying *L. paracasei* BL23-1E4 strain for oral administration to achieve protection against and treatment of GII.17 noroviral infections in healthy persons and immunocompromised patients of all ages. Because no standard animal model for human norovirus is available, we used a norovirus propagation system based on iPSC-derived human IECs to demonstrate that *L. paracasei* BL23-1E4 cells neutralized GII.17 (Kawasaki 308) norovirus. Because norovirus infection is associated with severe complications in infants, young children, and the elderly, a cold-chain-free lyophilized powder containing live bacteria, when mixed with a suitable excipient, may be useful for oral immunotherapy and prophylaxis against this pathogen. VHH-displaying *L. paracasei* BL23 represents an attractive approach for the prevention and treatment of norovirus infection in both developed and developing countries. 

## Figures and Tables

**Figure 1 pharmaceutics-15-00063-f001:**
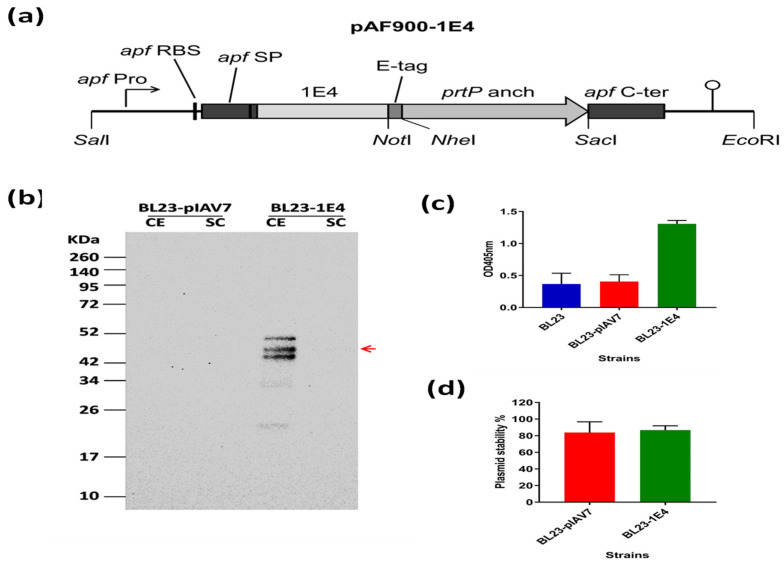
Production of VHH 1E4-displaying *Lactobacillus.* (**a**) Diagram of the expression cassette (pAF900-1E4) for cell-surface display of 1E4. VHH on *L. paracasei* BL23. *apf* Pro, the promoter of *apf* gene from *L. crispatus* M247; *apf* RBS, ribosomal binding site of *apf* gene; *apf* SP, the signal peptide of *apf* gene; *prtP* anchor, the 243 C-terminal amino acids of prtP from *L. paracasei* BL23; *apf* C-ter, C-terminal part of *apf* gene (not translated). The transcriptional terminator is indicated as the stem-and-loop structure. (**b**) Production of VHH 1E4 in cell extract (CE) and culture supernatant (SC) of cell wall-anchored recombinant *L. paracasei* BL23 strains tested by Western blotting using monoclonal mouse anti-E-tag antibody and horseradish peroxidase-labeled goat anti-mouse antibody as primary and secondary antibodies, respectively. *Lactobacillus paracasei* BL23 cells transformed with the empty vector pIAV7 were used as a negative control. The red arrow indicates the band corresponding to VHH 1E4 fused to the cell wall anchor region of PrtP. (**c**) Cell surface display of VHH 1E4 in recombinant *L. paracasei* BL23 strains according to ELISA assay using bacterial suspensions, a monoclonal mouse anti-E-tag antibody as the primary antibody, and alkaline phosphatase-conjugated rabbit anti-mouse antibody as the secondary antibody. Non-transformed *L. paracasei* BL23 cells and the strain transformed by the empty vector pIAV7 were used as negative controls. Signals were detected by reading the absorbance at 405 nm. The experiment was performed twice, and results are expressed as mean ± 1 standard deviation (SD). (**d**) Plasmid persistence of the recombinant *L. paracasei* BL23 strains expressing VHH 1E4 (BL23-1E4), measured according to colony count assays. The experiment was performed twice, and results are expressed as mean ± 1 SD. hpi, hours post-inoculation.

**Figure 2 pharmaceutics-15-00063-f002:**
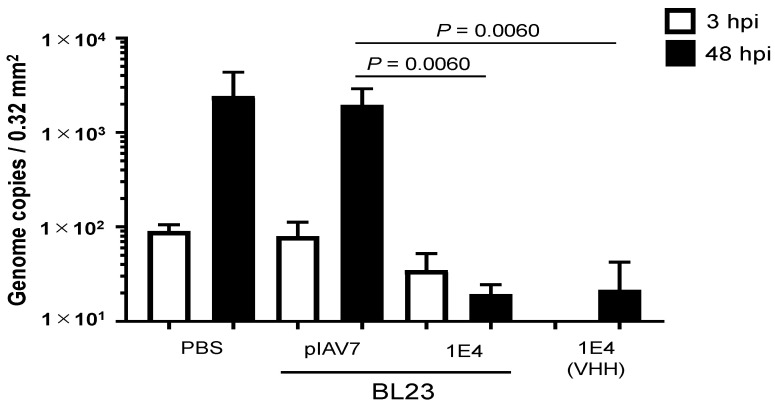
Neutralization of norovirus GII.17 in vitro by 1E4 nanobody-displaying *Lactobacillus* (BL23-IE4).We incubated 2 × 10^6^ genome equivalents of GII.17 (Kawasaki 308) human norovirus with 3 × 10^6^ cells of *L. paracasei* BL23-pIAV7 or BL23-1E4 or with 5 μg of recombinant VHH 1E4 for 2 h before using them to inoculate human intestinal epithelial cells. After inoculation, the cultures were incubated for 48 h in the presence of bile. Each data point is representative of at least three independent experiments, and data are shown as the mean ± 1 SD of six wells of supernatant from each culture group.

**Figure 3 pharmaceutics-15-00063-f003:**
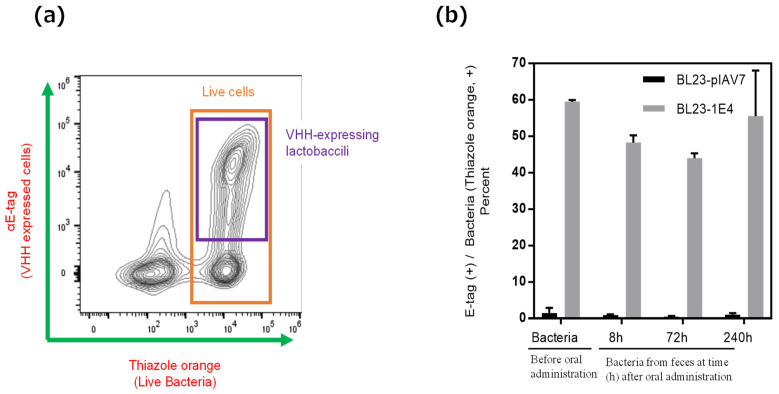
The expression of VHH 1E4 on the surface of lactobacilli collected from the feces of germ-free mice before and after oral administration with strain BL23-1E4. Flow cytometry was used to confirm the display of VHH on the surface of *Lactobacillus* strains. (**a**) Lactobacilli in the feces of germ-free mice before and after oral administration of *L. paracasei* BL23-pIAV7 or BL23-1E4 were stained with thiazole orange (TO), whereas 1E4 was detected via E-tag by using rabbit anti-E-tag IgG followed by brilliant violet 421-labelled goat anti-rabbit IgG. The purple box indicates the cell population that is double positive for TO and E-tag-associated 1E4. (**b**) The proportion (%) of E-tag-positive cells among TO-positive cells (i.e., live bacteria) is shown at various times (h) after oral administration of germ-free BALB/c mice with *L. paracasei* BL23-pIAV7 or BL23-1E4. The experiment was performed three times, and the results are shown as mean ± 1 SD.

**Figure 4 pharmaceutics-15-00063-f004:**
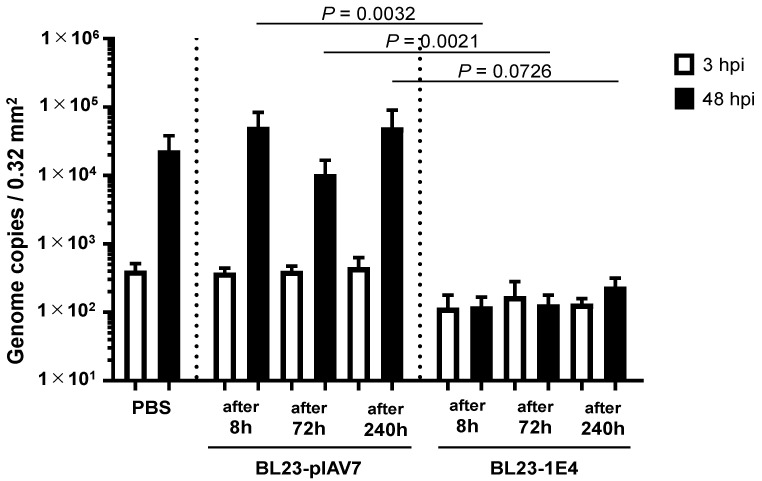
Neutralization of norovirus GII.17 in vitro by *Lactobacillus* BL23-IE4 that was collected from the feces of germ-free mice orally inoculated with 1E4 nanobody-displaying *Lactobacillus* (BL23-IE4). We incubated 2 × 10^6^ genome equivalents of GII.17 (Kawasaki308) human norovirus with 3 × 10^6^ cells of *L. paracasei* BL23-pIAV7 or BL23-1E4 for 2 h before using them to inoculate human intestinal epithelial cells. After inoculation, the cultures were incubated for 48 h in the presence of bile. Each data point is representative of at least 3 independent experiments, and data are shown as the mean ± 1 SD of 6 wells of supernatant from each culture group.

## Data Availability

Not applicable.

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
