# Peer review of "Lactobacilli as a Vector for Delivery of Nanobodies against Norovirus Infection"

_pharmaceutics, 2022, doi:10.3390/pharmaceutics15010063_

Round 1
Reviewer 1 Report
Manuscript ID: pharmaceutics-2069149
Lactobacillus expressing norovirus-specific nanobody for passive immunization against norovirus infection
Manuscript Type: Article
General Comments: This manuscript describes the construction, validation and in vivo persistence of recombinant Lactobacillus paracasei expressing a single domain antibody with neutralizing activity against GII.17 norovirus (VHH 1E4). The studies presented build on the existing body of work to exploit lactic acid bacteria as an inexpensive, logistically advantageous, orally-delivered preventative and therapeutic delivery platform. The manuscript is concise, well-written and provides a moderately innovative approach to address the human health threat posed by norovirus. Given the difficulty to test the intervention in a true challenge study, the authors have provided important proof-of-principal data that represents a valuable addition to the state of the art and the literature. The scientific premise is solid and the design and execution of the studies is appropriate. There are only a few minor comments and concerns provided below.
Specific Comments:
Line 40: It would more useful to identify key clinical aspects of norovirus that result in morbidity rather than the non-descript term of ‘brutal’.
Line 77: Please define ‘APF’.
Line 181: Correct ‘labbed’.
Line 202: Clarify what is meant by a ‘fluid based assay’. Is this an ELISA? This also needs to be clarified in the Figure 1C legend.
Line 233: It is surprising that the pH of the media was not affected. Are the bacteria able to metabolize/grow in the high oxygen conditions of the cell culture system?
Line 243: The title of this section is misleading as it implies that norovirus was neutralized in the host. What is shown is that bacteria recovered from a host still have the capacity to neutralize virus in an in vitro setting. Please clarify the title and the conclusions from this experiment.
Line 250/Figure S1: The figure is not particularly informative given the fogging of the plates. Please present colony counts over the days post inoculation rather than the photo. Clarify whether the germ-free mice eventually eliminate the L. paracasei altogether or whether the L. paracasei colonize the mice but lose the plasmid.
Line 278/Discussion: Please discuss how L. paracasei is likely to perform and persist in a host with a functional and diverse intestinal microbiome. Also discuss how the mechanism of norovirus neutralization by the recombinant bacteria in the in vitro system is similar or different to the expected mechanism in an infected host. Is this expected to be a prophylactic or therapeutic treatment?
Author Response
Reviewer 1
General Comments:
This manuscript describes the construction, validation and in vivo persistence of recombinant Lactobacillus paracasei expressing a single domain antibody with neutralizing activity against GII.17 norovirus (VHH 1E4). The studies presented build on the existing body of work to exploit lactic acid bacteria as an inexpensive, logistically advantageous, orally-delivered preventative and therapeutic delivery platform. The manuscript is concise, well-written and provides a moderately innovative approach to address the human health threat posed by norovirus. Given the difficulty to test the intervention in a true challenge study, the authors have provided important proof-of-principal data that represents a valuable addition to the state of the art and the literature. The scientific premise is solid and the design and execution of the studies is appropriate. There are only a few minor comments and concerns provided below.
Thanks for your positive comments.
Specific Comments:
Line 40: It would more useful to identify key clinical aspects of norovirus that result in morbidity rather than the non-descript term of ‘brutal’.
We have added the following text to the Introduction section of the revised manuscript (page 2, lines 41–43):
“Human norovirus causes an estimate 200,000 deaths annually worldwide and is a common infection in developed and developing countries in children younger than five years, the elderly, and immunocompromised people.”
Line 77: Please define ‘APF’.
We have defined this term—aggregation-promoting factor (page 3, line 77)
Line 181: Correct ‘labbed’.
We have corrected it (page 8, line 174) in the revised manuscript.
Line 202: Clarify what is meant by a ‘fluid based assay’. Is this an ELISA? This also needs to be clarified in the Figure 1C legend.
This technique follows the same principle as ELISA, but the bacteria are in suspension and not fixed on a plate. We have clarified this description (page 5, line 107).
In addition, we have revised the legend for Figure 1 (page 10, lines 212–213) to “Cell surface display of VHH 1E4 in recombinant L. paracasei BL23 strains according to ELISA assay using a bacterial suspension,…”
Line 233: It is surprising that the pH of the media was not affected. Are the bacteria able to metabolize/grow in the high oxygen conditions of the cell culture system?
GII.17 norovirus can survive in low pH (e.g., 2% citric acid, pH 3–5) and replicates in iPSC-derived IECs (Sato et.al. Sci Rep 2020: 10, reference 24). In fact, norovirus GII.17 replicates with and without L. paracasei BL23-pIAV7 under similar conditions (Figure 2). Although L. paracasei BL23 may be unable to grow under these assay conditions, 1E4 VHH–displaying L. paracasei were at least 1000 times more effective at neutralizing norovirus than was free VHH 1E4, as we discuss in our response to specific comment 3 from Reviewer 3.
We have added “ GII.17 Norovirus has the ability to survive in low pH (2 % citric acid, pH 3-5) and replicates in iPSC-derived IECs (Sato, S et.al. Sci Rep 2020, 10, reference 24).” (page 11, lines 226–227) in the revised manuscript.
Line 243: The title of this section is misleading as it implies that norovirus was neutralized in the host. What is shown is that bacteria recovered from a host still have the capacity to neutralize virus in an in vitro setting. Please clarify the title and the conclusions from this experiment.
We changed the section heading (page 11, lines 240–241) and revised the conclusion (page 12, line 253) in the revised manuscript:
In vitro norovirus neutralization by VHH-displaying Lactobacillus cells isolated from germ-free mice’s feces after their oral administration.
……and effectively neutralize norovirus in an in vitro assay, suggesting that VHH-displaying Lactobacillus are a potential anti-norovirus treatment.
Line 250/Figure S1: The figure is not particularly informative given the fogging of the plates. Please present colony counts over the days post inoculation rather than the photo. Clarify whether the germ-free mice eventually eliminate the L. paracasei altogether or whether the L. paracasei colonize the mice but lose the plasmid.
We now present the data previously shown in Figure S1 in Table S2 of the revised supplementary material.
Both BL23-PIAV7 and BL23-1E4 were present in feces by 240 h (10 days) after administration but were gone by 336 h (14 days). Orally administered lactobacilli, in general, transiently colonize the gastrointestinal tract for a maximum of approximately 1 month (Yue Y et al., Biomed Res Int 2020, 2020, reference 32) and Lactobacillus that displayed a VHH against rotavirus survived for only 2 to 4 days after oral treatment of wild-type mice (Pant N et al.,J Infect Dis, 2006, 194, 1580, reference 13). Therefore, we think that following oral administration in germ-free mice, L. paracasei BL23-1E4 will eventually be eliminated most likely due to loss of plasmid.
We have added this information to the Discussion section of the revised manuscript (page 17, lines 345–349).
Line 278/Discussion: Please discuss how L. paracasei is likely to perform and persist in a host with a functional and diverse intestinal microbiome. Also discuss how the mechanism of norovirus neutralization by the recombinant bacteria in the in vitro system is similar or different to the expected mechanism in an infected host. Is this expected to be a prophylactic or therapeutic treatment?
We added the following text to the Discussion section of the revised manuscript (page 16, lines 335–339):
“It is more likely that L. paracasei BL23-1E4 would be used as a prophylactic when outbreaks of norovirus infection occur, particularly during the winter season. However, lactobacilli are, in general, unlikely to persist long-term in the human intestine; for example, the probiotic strain L. rhamnosus GG remained for only about 1 week after oral administration was discontinued 12 days previously (Alander, M et.al.,Appl Environ Microbiol 1999, 65: 351, reference 31). Therefore, daily or weekly repeated oral administration of L. paracasei BL23-1E4 likely will be necessary, particularly during norovirus seasons.
Reviewer 2 Report
The article is well-written and very interesting. As in almost any manuscript, there are some typos here and there, but nothing important. I have almost no comments except for the following minor ones:
1) Use the same colors for the bars in Figures 1c and d.
2) What are all the bands in figure 1b? is that degradation?
3) In vitro propagation of noroviruses is extremely challenging. It would greatly benefit the readers if the authors could explain a little bit more about why they chose iPSC-derived IECs to propagate the virus. I understand they this team developed this system in a previous article, but it would enrich the text to add a few lines about this propagation system.
4) The first time VHH appears is in line 49. Please define there this acronym.
5) Please add the error bars to figues 1c and 2b. In fact, 2b is a representative data point of that experiment. Please plot a bar that has the mean value and the standard deviation of the three experiments.
Author Response
Reviewer 2
Comments and Suggestions for Authors
The article is well-written and very interesting. As in almost any manuscript, there are some typos here and there, but nothing important. I have almost no comments except for the following minor ones:
Thank you for your positive comments.
- Use the same colors for the bars in Figures 1c and d.
We have made this change in the revised manuscript.
- What are all the bands in figure 1b? is that degradation?
We now address this point in the revised manuscript (page 9, lines 192–194):
“The presence of higher molecular-weight bands is most likely due to peptidoglycan subunits covalently linked to the recombinant protein, which retard its migration, whereas the bands at lower molecular weights are degradation products within the bacterial cells due to overexpression of the recombinant protein (Navarre W, et.al. Microbiol Mol Biolo Rev 1999, 63, 174, reference 23)”.
A similar issue has previously been observed with overexpression of other recombinant proteins. We therefore have added reference 23 (Navarre W et.al. Microbiol Mol Biol Rev 1999, 63: 174) regarding the retarded migration of protein linked to peptidoglycan subunits to the revised manuscript.
- In vitro propagation of noroviruses is extremely challenging. It would greatly benefit the readers if the authors could explain a little bit more about why they chose iPSC-derived IECs to propagate the virus. I understand they this team developed this system in a previous article, but it would enrich the text to add a few lines about this propagation system.
We have added the following description to the Discussion section of the revised manuscript (page 15, lines 304–312):
“Human norovirus infects by attaching human histo-blood group antigens as co-receptors (Czako R et.al, Clin Vaccine Immunol 2012, 19, 284, reference 29); the primary receptor(s) for noroviral infection of host cells are unknown currently. Although human primary IECs, including iPSC-derived IECs, express histo-blood group antigens, norovirus replication typically also requires supplementation with bile, which contains unidentified components. In particular, GI.1, GII.3, and GII.17—but not GII.4—noroviruses require bile [16, 28]. Despite the use of bile, we think that the efficiency of norovirus replication in enteroid models including iPSC-derived human IECs is low compared with that of the in vivo human intestinal environment. Therefore, although the enteroid model does not completely mimic the human intestine, it remains effective as a neutralization assay.”
- The first time VHH appears is in line 49. Please define there this acronym.
We now define VHH—variable domain of llama heavy-chain antibody—in the Introduction section of the revised manuscript (page 2, line 49).
- Please add the error bars to figues 1c and 2b. In fact, 2b is a representative data point of that experiment. Please plot a bar that has the mean value and the standard deviation of the three experiments.
We have repeated the assay and added error bars to Figures 1c and 3b in the revised manuscript.

Reviewer 3 Report
My general comments:
1. The title in its present format may suggest that the methodology is fully established to tackle the problem of norovirus infection in human while this work is more in a developmental stage and is yet to be shown working in in vivo in primate animal models before going to human trials (see: Zhanlong He et al 2017; Emerg Infect Dis.; 23(2): 316–319). I suggest that the title to be modified to reflect this fact.
2. As authors elaborated about different genotypes of viruses in the GI and the fact that these genotypes prevalence have changed over time and the predominant genotype is now GII.17 in some part of Asia. However, the authors did not discuss anywhere in the manuscript that what would happen if GII.17 is switched with another genotype in the future and what kind of approach will be taken. Is it possible look for cross-reactive nanobodies recognizing most of the genotypes or have e.g., 3-5 nanobodies recognizing all the genotypes and then express them in lactobacillus separately and use the mix in the same assay reported in this manuscript?
3. I am aware that authors have used the iPSC-derived human IECs due to the lack of an animal model in their previous publication in 2021 and in this manuscript but there is little elaboration in the text if and how such an in vitro system would closely imitate the gut in vivo environment. A suitable non-human in vivo model will almost always be required before a human clinical trial and this needs to be elaborated.
4. A similar comment on the germ-free-mice and its compatibility with a real in vivo animal model need to be discussed. Would the Lacto-VHH bacteria behave the same if the gut microbiome with its highly complex and interactive micro-environment is there in vivo and will the outcome be different than what is observed in this manuscript.
5. I am wondering why author did not start with the VHH integration into the Lactobacilli genome approach knowing that antibiotic usage and plasmid stability are two main issues for the possible future application of the approach used in the manuscript. That would have been an added value to this work as the VHH-lactobacillus display/secretion has been out back in 2006 against other viruses (rotavirus) for which a mouse model system was also available. The fact that it is mentioned that authors have developed the genome editing tool (ref 26 and 27) and the genome integration of the VHH have advantages over the plasmid-based VHH expression, I wonder why authors did not use it in this study. Will there be an issue on the expression level when VHH is integrated into the genome?
My specific Comments:
1. It would have been best to use an anti-VHH antibody (from Jackson laboratory) in the WB to detect all the VHH products as E-tag may not showing the full picture of what has been expressed/degraded (Figure 1).
2. Would it possible for authors to quantify the display level of the VHH on lactobacillus and is the display method is more effective in neutralization of the virus than the soluble VHH (Figure 2). The fact that 5 ug VHH (approximately 2 x 1014 VHH molecules) and has a similar effect as 3 × 106 L. paracasei BL23-1E4. How this could be explained? Do we have an average of 2000 -6000 VHH/bacterial cell (as reported by Yin Lin, et al 2017 in Microbial Cell Factories)? What about the percentage of secreted VHH?
3. Spacing issue in line 180.
4. The fact that plasmid stability dropped 20% after 6-8 hr (line 208), shows that the plasmid approach used here could not be considered as a viable approach if this method is going to be used for the commercial application.
5. Is there any explanation for the drop in the norovirus genome copy number after 72 hr in figure 4 for the control BL23-pIAV7 and is this difference significant when compared with 8hr and 240 hrs?
6. I wonder if both reference 26 and 27 are required to be mentioned and either one would be sufficient.
Author Response
Reviewer 3
My general comments:
- The title in its present format may suggest that the methodology is fully established to tackle the problem of norovirus infection in human while this work is more in a developmental stage and is yet to be shown working in in vivo in primate animal models before going to human trials (see: Zhanlong He et al 2017; Emerg Infect Dis.; 23(2): 316–319). I suggest that the title to be modified to reflect this fact.
We understand that a few norovirus animal models, including a non-human primate model, are available, but no standard animal models for norovirus infection have been established so far. As described in our response to comment 3, the US Food and Drug Administration does not require in vivo efficacy testing in any animal model (including non-human primates) prior to human trials for the development of norovirus vaccine. Results from our enteroid model including iPSC-derived human IECs lead us to consider cell-wall–anchored VHH-lactobacilli as attractive oral nanobody delivery vectors for passive immunization against norovirus infection. Therefore, we have changed the title to:
“Lactobacilli as a vector for delivery of nanobodies against norovirus”
- As authors elaborated about different genotypes of viruses in the GI and the fact that these genotypes prevalence have changed over time and the predominant genotype is now GII.17 in some part of Asia. However, the authors did not discuss anywhere in the manuscript that what would happen if GII.17 is switched with another genotype in the future and what kind of approach will be taken. Is it possible look for cross-reactive nanobodies recognizing most of the genotypes or have e.g., 3-5 nanobodies recognizing all the genotypes and then express them in lactobacillus separately and use the mix in the same assay reported in this manuscript?
There is no universal VHH for neutralization among GII norovirus genotypes. For example, VHH 1E4 neutralizes GII.17 norovirus but not other GII genotypes (Yuki Yet al., J Infect Dis 2020, 222, 470, reference 18). Therefore, VHHs need to be developed for each genotype of noroviruses. We have added the following text to the Discussion section of the revised manuscript (page 15, lines 312–316):
“By using human IECs, we previously found that the cross-reactivity of VHHs against VLP of norovirus GII genotypes did not correlate with cross-neutralization activity and that there was no universal VHH for neutralization among GII norovirus genotypes. For example, VHH 1E4 neutralizes GII.17 norovirus but not do other GII genotypes [18]. Therefore, genotype-specific VHHs—including those for GII.2, GII.4, and GII.17 noroviruses—need to be developed.“
- I am aware that authors have used the iPSC-derived human IECs due to the lack of an animal model in their previous publication in 2021 and in this manuscript but there is little elaboration in the text if and how such an in vitro system would closely imitate the gut in vivo environment. A suitable non-human in vivo model will almost always be required before a human clinical trial and this needs to be elaborated.
We have added the following text to the Discussion section of the revised manuscript (page 15, lines 309–312):
“We think that the efficiency of norovirus replication in enteroid models including iPSC-derived human IECs is low compared with that of the in vivo human intestinal environment. Therefore, although the enteroid model does not completely mimic the human intestine, it remains effective as a neutralization assay.”
Although a few animal models of human norovirus infection have been reported, no standard animal model has been established so far. In terms of human trials, a GI.1/GII.4 norovirus systemic vaccine candidate (Takeda Vaccines) and a GI.I norovirus oral vaccine candidate (Vaxart) have entered Phase II and Phase I testing, respectively, without prior evaluation in animal models, including non-human primates (references 4–6 in the revised manuscript). These products have been tested only through in vitro inhibition of HBGA carbohydrate binding, which correlates with neutralization of norovirus GII.4 but which is not a neutralization assay. Regardless, the US Food and Drug Administration does not require efficacy testing in a non-human primate model for the development of norovirus vaccine.
- A similar comment on the germ-free-mice and its compatibility with a real in vivo animal model need to be discussed. Would the Lacto-VHH bacteria behave the same if the gut microbiome with its highly complex and interactive micro-environment is there in vivo and will the outcome be different than what is observed in this manuscript.
We understand that the iPSC-derived human IECs model does not completely mimic the human intestine, as we discussed in our response to comment 3. We now address this point in the Discussion section of the revised manuscript (page 15, lines 309–312). In addition, due to competition with other bacteria in the complex intestinal environment, lacto-VHH bacteria will need to be administered regularly (see page 16, lines 335–339).
- I am wondering why author did not start with the VHH integration into the Lactobacilli genome approach knowing that antibiotic usage and plasmid stability are two main issues for the possible future application of the approach used in the manuscript. That would have been an added value to this work as the VHH-lactobacillus display/secretion has been out back in 2006 against other viruses (rotavirus) for which a mouse model system was also available. The fact that it is mentioned that authors have developed the genome editing tool (ref 26 and 27) and the genome integration of the VHH have advantages over the plasmid-based VHH expression, I wonder why authors did not use it in this study. Will there be an issue on the expression level when VHH is integrated into the genome?
We prefer to test first the function of Lactobacillus displayed VHH using the plasmid expression system as the integration of VHH is more time-consuming. Now that the function of the VHH has been demonstrated, we will integrate the expression cassette.
We are not expecting an issue of expression when the VHH is integrated on the chromosome as the overexpression of the VHH-prtP fusion protein using the plasmid system saturate the secretion and anchoring machinery of lactobacilli. We previously compared the activity of L. paracasei producing surface-anchored ARP1 (VHH targeting rotavirus) using the plasmid (L. paracasei pAF900-ARP1) and integration system (L. paracasei EM233) in vitro and in vivo. When the display of the ARP1 on the surface of bacteria was evaluated by flow cytometry, the fluorescence intensity was shown to be 6 times lower for L. paracasei EM233 than for the corresponding plasmid construct, L. paracasei pAF900-ARP1 (Marin MC et al. Viruses,2019, 11, reference 26). The reduction in antibody display is acceptable considering that each bacterium contains one copy of the chromosome but multiple copies of the plasmid (n = 162). Furthermore, the binding of whole cells of L. paracasei EM233 and L. paracasei pAF900-ARP1 to rotavirus was shown to be similar using flow cytometry. Most important, the level of ARP1 antibody fragments displayed in L. paracasei EM233 was sufficient to reduce infection when tested in an animal model of rotavirus infection and to the same level as L. paracasei pAF900-ARP1 (Marin MC et al. Viruses,2019, 11, reference 26).
We added the following sentence to the revised manuscript (page 14, lines 295–298):
“We previously showed that. L. paracasei BL23 producing surface-anchored ARP1, engineered using either a plasmid or integration system, conferred similar protection in mouse model of rotavirus suggesting the feasibility of using a chromosomal integrated expression system for delivery of VHH against norovirus (Marin MC et al. Viruses,2019, 11, reference 26).”
My specific Comments:
- It would have been best to use an anti-VHH antibody (from Jackson laboratory) in the WB to detect all the VHH products as E-tag may not showing the full picture of what has been expressed/degraded (Figure 1).
We agree that using an anti-VHH would give a better idea of the amount of degraded VHH products but would be less indicative of the amount of intact protein (of the correct molecular weight) expressed on the surface. We do not expect much degradation when we express VHH at lower levels by using the chromosomal integration system (Marin MC et al. Viruses,2019, 11, reference 26).
- Would it possible for authors to quantify the display level of the VHH on lactobacillus and is the display method is more effective in neutralization of the virus than the soluble VHH (Figure 2). The fact that 5 ug VHH (approximately 2 x 1014 VHH molecules) and has a similar effect as 3 × 106 L. paracasei BL23-1E4. How this could be explained? Do we have an average of 2000 -6000 VHH/bacterial cell (as reported by Yin Lin, et al 2017 in Microbial Cell Factories)? What about the percentage of secreted VHH?
According to the reviewer’s request, we have added the following text to the Discussion section of the revised manuscript (pages 15–16, lines 318–326).
“According to flow cytometry using calibrated fluorescent microspheres, roughly 1000 VHH were displayed on the surface of each bacterial cell. Thus, 3 × 106 L. paracasei BL23-1E4, which completely neutralized 2 × 106 genome equivalents of norovirus GII.17, contains approximately 3 × 109 VHH molecules on the cell surface. According to our previous paper [18], 0.05 to 5 mg of VHH 1E4 inhibited 2 × 106 genome equivalents of norovirus GII.17, yielding approximately 2 × 1012 to 2 × 1014 VHH molecules (Lin Y et.al Microbiol Cell Fact 2017, 16, 1 reference 30). Therefore, 1E4 VHH–displaying L. paracasei was at least 1000 times more effective at neutralizing norovirus than was free VHH 1E4. The numerous antibody fragments expressed on the bacterial surface result in the formation of ‘biological beads’ that afforded high-avidity binding due to multivalency and, thus, promoted strong agglutination and subsequent neutralization of the virus.”
- Spacing issue in line 180.
We have deleted the extraneous space from the revised manuscript.
- The fact that plasmid stability dropped 20% after 6-8 hr (line 208), shows that the plasmid approach used here could not be considered as a viable approach if this method is going to be used for the commercial application.
We think that plasmid-based recombinant BL23 1E4 is not clinically applicable, not only because of plasmid instability but also environmental contamination. We therefore intend to produce lactobacillus displaying norovirus-specific nanobody by using chromosomal integration of the expression cassette and marker-free selection (see pages 16–17, lines 339–354).
- Is there any explanation for the drop in the norovirus genome copy number after 72 hr in figure 4 for the control BL23-pIAV7 and is this difference significant when compared with 8hr and 240 hrs?
We do not know, but there is a significant difference between 8 and 240 h for BL23 pIAV7 as the control group (t-test, P = 0.036) in these experiments but no difference between these time points for BL23 1E4 (t-test, P = 0.639).
- I wonder if both reference 26 and 27 are required to be mentioned and either one would be sufficient.
We deleted reference 26 in the original submission from the revised manuscript.
